# A Comparison of Diet Quality in a Sample of Rural and Urban Australian Adults

**DOI:** 10.3390/nu13114130

**Published:** 2021-11-18

**Authors:** Rebekah Pullen, Katherine Kent, Matthew J. Sharman, Tracy L. Schumacher, Leanne J. Brown

**Affiliations:** 1School of Health Sciences, University of Tasmania, Launceston, TAS 7250, Australia; rebekah.pullen@utas.edu.au (R.P.); matt.sharman@utas.edu.au (M.J.S.); 2School of Health Sciences, Western Sydney University, Campbelltown, NSW 2560, Australia; 3Department of Rural Health, The University of Newcastle, Tamworth, NSW 2340, Australia; 4Hunter Medical Research Institute, New Lambton Heights, NSW 2305, Australia

**Keywords:** Australian dietary guidelines, Australian recommended food score, diet quality, diet variety, rural

## Abstract

The diet quality of rural Australians is under researched. Characterising disparities in diet quality between rural and urban populations may inform targeted interventions in at- risk groups. A cross-sectional study aimed to determine the relationship between diet quality, rurality and sociodemographic characteristics in a sample of Australian adults. Participants were recruited at rural and regional events between 2017 and 2020, in New South Wales, Australia. Diet quality was measured using the Healthy Eating Quiz or Australian Eating Survey to generate an Australian Recommended Food Score (ARFS). ARFS was compared by rurality and sociodemographic characteristics using multivariate regression. Participants (*n* = 247; 53% female) had a mean ± SD ARFS of 34.5 ± 9.0. There was no significant effect of rurality on ARFS (β-coefficient = −0.4; 95%CI −3.0, 2.3). Compared to participants aged 18–30 years, higher ARFS was evident for those aged 31–50 (β = 5.4; 95%CI 0.3, 10.4), 51–70 (β = 4.4; 95%CI 0.3, 8.5) and >71 years (β = 6.5; 95% CI 1.6–11.4). Compared to those living alone, participants living with a partner (β = 5.2; 95%CI 2.0, 8.4) and families with children (β = 5.6; 95%CI 1.4, 9.8) had significantly higher ARFS. ARFS was significantly lower with each additional self-reported chronic health condition (β = −1.4; 95%CI −2.3, −0.4). Our results indicate that diet quality as defined by the ARFS was classified as ‘getting there’ and that age, living arrangements and chronic health conditions, but not rurality, influenced diet quality in a sample of Australian adults.

## 1. Introduction

Chronic diseases such as heart disease, stroke and type 2 diabetes are currently the leading cause of mortality and morbidity worldwide and have been described as the greatest public health challenge of the 21st century [1]. In 2018, 38% of the burden of disease in Australia was attributed to modifiable risk factors, with dietary risk factors directly accounting for 5.4% of the total burden of disease [2]. Worldwide, specific dietary risk factors for the development of diet-related disease have been identified as low consumption of fruits, vegetables and wholegrains, together with rising consumption of sodium and saturated fats [3,4,5,6,7,8].

In 2018, approximately 29% of the Australian population lived in rural or remote locations [9]. People living in rural and remote locations have shorter lives and experience disproportionately higher levels of chronic diet-related disease such as coronary heart disease, stroke, chronic kidney disease and type 2 diabetes when compared to their metropolitan counterparts [9,10,11,12]. Rural Australians experience many sociodemographic precursors to poor diet quality such as low education, low income and high unemployment [11,13]. Other factors that influence health outcomes in rural settings are location specific, including reduced access to health services combined with lower health literacy, and the relatively higher cost of, and poorer access to, fresh foods [14,15,16,17].

Evidence has demonstrated that components of food work synergistically, influencing the risk of developing chronic diseases [18,19]. As a consequence, nutrition research has moved away from the traditional approach of investigating single nutrient or specific foods when investigating diet and its relationship to disease towards analysing the patterns in which foods are consumed in a whole-of-diet context [18,20,21]. Dietary patterns research describes the consumption of nutrients, foods and food groups as well as examining the variation, diversity, frequency and quality of foods in the diet [20]. Recent definitions of diet quality incorporate the concepts of diversity, adequacy, proportionality and moderation, with the primary objective being a balance between each [21,22,23]. These concepts have informed the Australian Dietary Guidelines (ADG) and accompanying resources such as the Australian Guide to Healthy Eating (AGHE). The ADG identifies an evidence-based, optimal dietary pattern and presents the types and amounts of foods that Australians are recommended to eat for health and wellbeing. It consists of five core food groups representing similar nutrients within each, and advocates that a variety of foods should be consumed from the five food groups each day. One measure of diet quality is compliance with the ADG, which has been shown to reduce risk factors associated with the development of chronic diseases such as cardiovascular disease, obesity and hypertension [21,24,25].

Diet quality can be measured using two approaches: data-driven posteriori methods or a priori methods such as dietary quality indices (DQI) [8,21,26]. Higher DQI scores reflect closer adherence to dietary guidelines and therefore higher diet quality. In Australia, Indigenous Australians, those living with disabilities, the unemployed, single-parent households and people living in rural and remote communities have been identified as particularly vulnerable to poor diet quality and therefore at increased risk of chronic diet-related disease [9]. Investigations into diet quality of rural and remote Australians remains under prioritised [12]. For example, between 2000 and 2014, 184 (1.1%) of the total 16,651 National Health and Medical Research Council (NHMRC)-funded projects were defined as relating to Australian rural health research [12]. This, together with methodological challenges, has led to the under representation of these populations in dietary research, and has meant that the full breadth and interaction between determinants affecting dietary status have yet to be fully characterised [11,12].

Despite the under representation of rural and remote-dwelling adults, Australian national survey data identified a trend consistent across rural areas, with only 1 in 10 individuals meeting the ADG recommendations for fruits and vegetables [11], a further indication of the need for research and intervention. In order to prioritise this population in public health policy, studies investigating the sociodemographic and dietary characteristics of this population are urgently required. Therefore, the aim of this study is to determine the relationship between diet quality, rurality and sociodemographic characteristics in a sample of Australian adults recruited from rural and regional events.

## 2. Materials and Methods

The ‘Changing Health Actions at Rural and reGional Events in 20 minutes’ (CHAaRGE:20) project was conducted to determine the health status of participants attending rural and regional events using opportunistic face-to-face engagement in health-related activities. Ethics approval was obtained from the University of Newcastle Human Research Ethics Committee (H2017-10979).

### 2.1. Participants and Recruitment

Between 2017 and 2020, participants were recruited from Tamworth Country Music Festival (TCMF) (Tamworth, NSW, Australia) and AgQuip Field Day (AgQuip) (Gunnedah, NSW, Australia). These events were selected due to a high proportion of attendees being from rural locations. Participants were eligible for participation if they were over the age of 18 years and were not under the influence of drugs or alcohol. They were required to have proficient use of English and basic literacy level. Recruitment was achieved through convenience sampling methods such as volunteers handing out flyers, banner advertisements, media and word of mouth. Interested participants were provided a verbal and written summary outlining the project aims, specific measures being collected and participant-required tasks. Participants were offered individualised feedback about their health assessment results from a qualified dietitian. Written consent was obtained for all participants.

### 2.2. Measures

Participants answered survey questions relating to demographics and self-reported health conditions. Anthropometric measures were taken by trained researchers or student volunteers. The Healthy Eating Quiz (HEQ) [27] was completed on a tablet or laptop either before or after anthropometric measures to optimise waiting time in 2017. Given the time burden for answering the HEQ, this option was changed to the voluntary completion of the Australian Eating Survey (AES) [28] in January 2018. This survey was disseminated via email and completed in participants’ own time, following the event. Study data were stored using REDCap electronic data capture, hosted at Hunter Medical Research Institute [29,30]. AES was collected by SurveyMonkey for the TCMF in 2018, prior to being collected by REDCap for AgQuip in 2018.

### 2.3. Dietary Assessment

The Australian Recommended Food Score (ARFS) is a validated dietary index modelled on diet variety from within and between food groups, as described in the Australian Dietary Guidelines (ADG) [28,31,32,33]. The ADG has five core food groups, which are organised according to their similar nutrient profiles. The HEQ consists of a validated subset of 70 questions from which the ARFS is derived [28,33]. Eight subscales within the HEQ reflect the core food groups of the ADG. The subscales are composed of 20 questions dedicated to vegetables, 12 to fruit, 7 to meat, 6 to plant-based protein foods, 12 to breads and cereals, 10 to dairy foods and 1 to water [33]. Calculation of relevant points from each subscale provided a total ARFS score (range 0–73) and subscale scores for diet variety. An ARFS score can be categorised into four groups: ‘needs work’ (<33), ‘getting there’ (33–38), ‘excellent’ (39–46) or ’outstanding’ (47+). However, very high ARFS may indicate energy intake that is excessive to need. The ARFS has been demonstrated to be a reliable and valid measure of diet quality [28,32]. The higher the score on the ARFS, the greater the variety of nutrient-dense foods consumed and therefore the greater the diet quality [34]. The ARFS is also able to be derived from the Australian Eating Survey (AES), a more comprehensive food frequency questionnaire, which is inclusive of other foods, including discretionary foods [32].

### 2.4. Anthropometric Characteristics

Height and weight were measured using a Biospace BSM370 Automatic BMI Scale Stadiometer. Where multiple height and weight values were present, they were summed and divided for an average to enhance accuracy. BMI was calculated from measurements of participants’ height and weight (weight (kg)/height (m^2^), then categorised into “normal weight” (18.5–24.99 kg/m^2^), “overweight” (25.0–29.99 kg/m^2^) and “obese” (Obese I: 30.0 to 34.99 kg/m^2^, Obese II: 35.0 to 39.99 kg/m^2^, Obese III: 40.0 kg/m^2^), as defined by the World Health Organisation [35].

### 2.5. Sociodemographic Characteristics

Participants self-reported living circumstances, household income, highest level of education completed, smoking status and diagnosed chronic health conditions. Participants reported having diagnosed chronic health conditions by answering the question: “*Have you EVER been told by a doctor or health professional that you have any of these conditions? Tick any that apply*”. Respondents selected any appropriate chronic condition from a list of predefined conditions that have been associated in the literature to be prevalent in rural populations [9,11]. The variables of age education, household income and living arrangements were collapsed due to low cell counts for data analysis. Age categories were reduced from sixteen levels (18–25; 26–30; 31–35; 36–40; 41–45; 46–50; 51–55; 56–60; 61–65; 66–70; 71–75; 75–80; 81–85; older than 85; under 18 years; don’t wish to answer) to four (18–30; 31–50; 51–70; >71). Education categories were reduced from eight levels (less than year 10, Year 10 or 11, Year 12; Trade or Vocation; University or graduate degree; Postgraduate degree or higher; None of the above; I don’t wish to answer) to four (Year 12 and less; Certificate or Diploma; University). Household income categories were reduced from fourteen levels (No in-come; Pension; AUD 1–6293; AUD 6240–15,999; AUD 16,000–25,999; AUD 26,000–36,399; AUD 36,400–51,999; AUD 52,000–77,999; AUD 78,000–103,999; AUD 104,000–129,999; AUD 130,000–155,000; >AUD 156,000; I do not know; I do not wish to answer) to six (No income; Pension; AUD 1–51,999; AUD 52,000–103,000; >AUD 104,000; I don’t know). Living arrangements categories was collapsed from seven levels (living alone; partner/spouse; own children; someone else’s children; parents; other adults; I don’t wish to answer) to four (living alone; partner/spouse only; single/partnered with children; parents and other).

Self-reported chronic health conditions for each individual were summed and used to calculate the average number of chronic health conditions for the sample. Individual conditions with related aetiology were listed together to form an overarching condition classification. Circulatory conditions incorporated cardiovascular disease (CVD), heart disease (HD), high blood pressure (HTN) and high blood cholesterol. Chronic mental health included anxiety, depression, schizophrenia and any other diagnosed mental health condition. Musculoskeletal conditions included back problems, osteoporosis and rheumatoid arthritis and any other related diagnosed conditions. Respiratory conditions included asthma, chronic obstructive pulmonary disease COPD and any other diagnosed lung-related condition. Rurality was calculated according to ARIA+ scores [36] using post codes into major cities, inner regional, outer regional, remote and very remote categories. These categories were then collapsed into major cities and regional and remote. ARIA+ was used as it is a recognised standard measure of rurality. It was selected over other measures due to its sensitivity and is considered to be the most stable measure over time as it is based on road distance travelled for locality to service centres rather than factors of population density [36,37]. The use of ARIA+ allows for the capacity to conduct comparisons between population-based Australian Bureau of Statistics survey results and present and future variations in this study.

### 2.6. Statistical Analysis

Statistical analysis was conducted using IBM SPSS Statistics for Windows, version 26.0 (IBM Corp. Armonk, NY, USA). Power calculations were conducted [38]. A true detectable difference of 3.48 in ARFS can be found in a sample of 246 participants, based on a standard deviation of 9.7 [33], an alpha value of 0.5 and a beta of 0.8, and 1:1 ratio in each group. The demographic characteristics of the included sample were compared to those excluded from analysis to determine whether those providing dietary data were reflective of the whole cohort, using Chi Square tests. Continuous data were tested for normal distribution by visual methods of a histogram, quantile-quantile plot (QQ-plot) and formally tested using Shapiro–Wilk. Any variables that had *p* values greater than *p* = 0.05 in the Shapiro–Wilk test were reported as median and interquartile range. Data that were normally distributed were reported as means and standard deviations. Univariate tests assessed the differences in mean ARFS scores according to levels within sociodemographic variables. For the multivariate regression, respondents nominating survey answers *“I don’t know”* or *“I don’t wish to answer”* for variables with this option were coded as missing. Assumptions of linearity, homoscedasticity and independence were tested visually by means of scatter plots and box plots. Two outliers were identified; however, they were retained, as both outlier ARFS were within a justifiable range, and sensitivity tests indicated no change in univariate and multivariate linear regression models. All variables included in the analysis satisfied the assumption of equal variance, demonstrating properties of homogeneity. The significance level for inclusion of those variables investigated in univariate regression in the further multivariate linear regression model was set to *p* ≤ 0.20. Rurality was included in the regression (regardless of *p* value) due to the relationship with the research question and strong links in the literature to diet variety [10,16,39,40,41].

## 3. Results

In this study, 638 participants were surveyed between 2018 and 2020. Of these, 391 participants had missing total and/or subscale ARFS and were therefore excluded. Participants who provided complete dietary data allowing a total ARFS to be calculated were significantly more likely to be female, with an income of up to AUD 51,000, aged between 51 and 70 years of age and have an education level of year 12 or under. Sociodemographic characteristics of the study sample are reported in Table 1. Within the total sample, the majority were female (52.6%) and lived in regional and remote locations (77.9%), with a large proportion classified as overweight (39.6%). The largest proportion of study participants had an education level of year 12 and under (45.7%), and 30.4% of participants reported income between AUD 1–51,999. Most (52.0%) lived with a spouse/partner only, while 17.5% lived with a partner/spouse and children. Of the total sample, 29.1% had previously smoked.

Diagnosed chronic health conditions by gender are reported in Table 2. The most prevalent chronic health conditions were circulatory conditions (27.7%), musculoskeletal conditions (23.7%) and overweight and obesity (21.2%) overall. For males, the most prevalent conditions were circulatory conditions (26.0%), overweight and obesity (21.7%) and musculoskeletal conditions (22.6%). The data highlight that more than a quarter of women in this sample had circulatory conditions (27.6%), musculoskeletal conditions (24.6%), were overweight or obese (20.8%) or had chronic mental health conditions (18.5%), which was higher compared to males. The average number of summed chronic health conditions for the total sample was 1.2 (±1.2), which was similar for both males and females.

Total ARFS and ARFS subscales are reported in Table 3. Overall, the mean total ARFS was classified as “*getting there*” (34.5%) (Table 3). Females reported higher (35.5) mean total ARFS compared to males (33.4) (Table 3). When compared to males, females reported higher ARFS in subscale categories of vegetables (13.9), fruit (5.7), meat alternatives (2.3), grains (5.1), dairy (4.1) and water (0.7) (Table 3). Males reported higher subscale ARFS for the categories of meat (3.2) and extras (0.9) (Table 3). The ARFS for the total sample, as well as by gender in all subscale categories, except for vegetables, extras and water, was reported as being below half of their associated maximum score (Table 3).

Results of the multivariate linear regression for the effect of sociodemographic categories on total ARFS are reported in Table 4. Age, living arrangements and number of health conditions were retained for multiple regression analysis due to *p* < 0.20. No effect of rurality on ARFS was found in the multivariate analysis. Significant differences in total ARFS for all levels of age were found when compared to the reference group of 18–30 years old. The largest difference was detected for the 31–50 and 71 > age groups with a difference in ARFS of 5.4; *p* = 0.037 and 6.5; *p* = 0.010, respectively. Mean differences in ARFS were detected between all levels of living arrangements, demonstrating increased total ARFS for all levels when compared to the reference group living alone (Table 4). For every additional diagnosed health condition, the ARFS was reduced by −1.4; *p* = 0.004. The final model had an adjusted R square of 0.077, *p* < 0.001.

## 4. Discussion

This study investigated the relationship between sociodemographic characteristics and diet quality using the Australian Recommended Food Score (ARFS) in a sample of rural and urban Australian adults. The average ARFS was categorised as “*getting there*”, indicating a need to improve diet quality. Rurality had no significant effect on diet quality in this sample. Analysis of other sociodemographic characteristics showed living arrangements and the number of diagnosed chronic health conditions had the strongest associations with diet quality, where living alone and having multiple chronic health conditions were associated with poorer diet quality. Our study examined many of the same sociodemographic characteristics as previous studies [28,33,39,41,42,43,44,45,46,47,48,49] and observed similar findings showing a lack of relationship between diet quality, gender, socioeconomic status and education.

The literature has shown that adherence to the ADG within the Australian population is poor, resulting in low diet quality [45,50,51]. The mean diet quality score in our study (34.5 ± 9.0) was comparable with another study of Australian adults which found a mean total diet score of 34.1 using the ARFS from the HEQ [33]. Further, two Australian studies also with adult samples using the ARFS from the AES found a mean diet quality score of 33 ± 8.8 [32] and median score of 36 [28]. In this study, the diet quality score of “*getting there*” was driven by low individual scores for all the ARFS subscales. This finding is consistent with both national and international studies across a variety of diet quality indices, which show that low diet quality is driven by low scores across most, if not all, food groups [33,40,41,42,43,44,45,48,52,53].

While our study did not demonstrate a relationship between diet quality in rural and urban dwelling adults, this may be due to the limited sample size of our study. However, some literature has demonstrated that rural populations experience many of the sociodemographic precursors to poor diet quality such as low income, low education attainment and limited access to fresh foods, which could negatively affect diet quality [11,16,54]. Importantly, two recent systematic reviews highlight the scarcity of studies surrounding rural populations [6,55], limiting comparisons to our data. In a sample of older Australian adults using the DGI-2013 (score range 0–130), one study found that men but not women from rural areas had significantly lower total diet quality scores (80.1) compared to their urban counterparts (83.0). The authors concluded that rural-related disadvantage was the mediator of poor diet quality scores [41]. A second study in women of reproductive age found no difference between total diet quality scores between urban (84.8) and rural (83.9) women using the DGI. Unlike the present study, it found that rural women had a significantly higher component score for meat and meat alternatives [55]. The inconsistent effect of rurality on diet quality indicates that differences in diet quality may be driven by differences in rural food environments in different regions and, therefore, that multi-site trials comparing multiple rural populations should be a consideration for future research.

There are only a few Australian studies investigating living arrangements as a sociodemographic variable influencing diet quality, particularly in rural populations [33,56,57,58,59]. Studies investigating the effect of living arrangements of diet quality have focused predominantly on ageing populations. An international systematic review of the relationship between living arrangements and diet found that a number of studies consistently identified that living alone led to lower fruit and vegetable intake and lower adherence to dietary recommendations [60]. The determinants of low diet quality in those living alone are complex. Demographic characteristics influence the likelihood of a person living alone, including gender, socio-economic status and age, which may influence the relationship between living alone and low diet quality [60]. Psychosocial drivers of poor diet quality may affect those living alone, including decreased motivation and enjoyment of cooking, which increases the consumption of pre-packaged processed meals that are high in sodium, sugar and trans/saturated fats [61,62]. A lack of support for and encouragement for maintaining adherence with dietary recommendations has also been recognised as a difference experienced between those living alone and those living with others [63]. Similarly, one study demonstrated that those living in arrangements reflecting cohabitation or marriage-like arrangements have a higher adherence to dietary recommendations; this could be due to more regular and formalised shopping and eating habits [60,64,65], as well as social facilitation [63]. An Australian study exploring sociodemographic characteristics on diet quality in a sample of adults responding to the HEQ, incorporated a measure of meal sharing as a possible determinant of diet quality [33]. The results support the present study’s finding that, when controlling for age, sex and socioeconomic status, diet quality increases in line with the number of people whom a person shares meals with as opposed to those who eat meals alone [33].

Given the association between diet and disease, and considering the increased prevalence of multimorbidity [10], it is surprising that there are so few studies, particularly in rural populations, investigating the effect of the presence of chronic health conditions on diet quality. In the present study, we found that, for every additional diagnosed chronic health condition in an individual, total diet quality score decreased. Previous research indicates that even small decreases in total diet quality scores are associated with increased all-cause and specific-cause mortality. For example, an American study [66] found that an increase in diet quality score up to or above the 20th percentile in their population group over 12 years was significantly associated with a reduction in total mortality of between 8 and 17%, as well as significantly lowering the risk of death from cardiovascular disease. In contrast, their results also suggested that decreases in diet quality over a 12-year timeframe, when compared to no change, were associated with an increase in total mortality between 6 and 12% [66]. Interestingly, the study suggests that an increase of 22 of the 110 available points within the Alternative Healthy Eating Index (AHEI) over a 12-year period could reduce risk of death by 20%, which could be achieved by increasing intake of nuts and legumes from 0 to 1 serving per day and reducing red or processed meats by 1.5 servings per day [66]. An Australian study [25] using the Total Diet Score (TDS) (range 0–20) found that those who had diets that closely adhered to the Australian Dietary Guidelines, as reflected by higher diet quality scores, had a 21% reduced risk of all-cause mortality and 23% decreased risk of cardiovascular mortality. This study found that, with every increase in the standard deviation of TDS (1 SD = 2.19 units of TDS), there was an 8% decrease in risk of all-cause mortality [25]. Further to this, another American study [67] exploring the relationship between four disease risk factors on diet quality found that those with one or none of the clinical risk factors had significantly higher total diet quality score (55.7 out of a possible 100), as measured by the Healthy Eating Index-2015, when compared with those who had all four risk factors. Those with all four risk factors had a significantly lower diet quality score of (51.1). Australian studies have explored the relationship between health and diet in terms of specific disease outcomes, mortality and relative risk or self-reported perception of health rather than the number of diagnosed chronic health conditions [45,46]. Studies that have measured the effect of perceived health status on diet quality found those who had a greater self-perceived health status demonstrated higher total diet quality scores [45,49].

A major strength of the present study was the use of validated food frequency methods in the assessment of dietary intake, which are able to accurately capture usual dietary intake, including temporal changes in eating patterns; as this is a limitation of the existing literature on dietary intakes in rural populations [54]. Additionally, relative to the published literature, there was a strong representation of rural and regional participants, demonstrating that engaging rural adults in settings where they work, live and play is key to increasing participation rates of rural people in research activities. Previously, the literature has identified this population as underrepresented and hard to access, and as such, recruitment at rural and regional events should be considered as a strategy for future research. TCMF and AgQuip provide examples of annual rural and regional events that are important pillars of rural and regional communities. These events are hosted at local community venues and possible to attend free of charge, which supports their use in research aiming to sample a diverse rural and regional population. However, the limitations of this include potentially under-representing some groups who do not attend these events. For example, our study may have over-represented the ‘walking well’ who are able to attend community-based health promotion activities and who have an interest in health or nutrition, meaning that those with chronic conditions and poor diet may have been underrepresented. Supporting this, our study found significant differences in age, gender, income and education within the CHAaRGE:20 study sample for those people who completed the comprehensive dietary assessment versus those who did not, potentially limiting the generalisability of our results. While successful in recruiting a high proportion of rural and regional-dwelling participants, our study also demonstrates the complexity of sampling in rural and remote communities and presents some important considerations for future diet quality studies. Another limitation of our study is that it was likely underpowered in detecting an effect of rurality on diet quality in this sample, as recruiting was suspended due to the COVID-19 pandemic. Further, in our study, there was a high prevalence (21.1%) of respondents not wishing to disclose income. This could have produced misleading results and ignored the contribution of income in diet quality. Further research in a larger sample size from multiple rural and remote regions would support stratifying diet quality results according to all levels of rurality (regional, rural, remote and very remote populations), which would enhance the sensitivity of our analyses. Additionally, future research may consider not grouping health conditions into predefined groups, so a more thorough exploration between diet quality and health outcomes can be prioritised.

## 5. Conclusions

This present study suggested that the diet quality in our sample of rural and urban Australian adults was classified as “*getting there*”, and that rurality did not influence diet quality in this sample. It highlighted that living alone may be a primary driver of diet quality and this sociodemographic characteristic should be investigated further in future research. Additionally, multimorbidity was associated with reduced diet quality, suggesting that efforts must be made to assist Australian adults living with chronic diseases to improve their diet quality, potentially improving their health outcomes. Further studies of diet quality that specifically represent the diverse experiences of Australians living in rural or remote locations are required.

## Figures and Tables

**Table 1 nutrients-13-04130-t001:** Total number and percentage for sociodemographic characteristics in a sample of Australian adults.

		Male		Female		Total	
		*n* = 117		*n* = 130		*n* = 247	
Characteristics	Levels	*n* (100%)		*n* (100%)		*n* (100%)	
Age (years)	18–30	20	(17.1%)	24	(18.5%)	44	(17.8%)
	31–50	16	(13.7%)	24	(18.5%)	40	(16.2%)
	51–70	57	(48.7%)	68	(52.3%)	125	(50.6%)
	>71	24	(20.5%)	14	(10.8%)	38	(15.4%)
Rurality	Major cities	27	(23.1%)	30	(23.1%)	57	(23.1%)
	Regional/remote	90	(76.9%)	100	(76.9%)	190	(77.9%)
Education	≤Year 12	53 ^1^	(45.7%) ^1^	59 ^2^	(45.7%) ^2^	112 ^3^	(45.7%) ^3^
	Cert ^a^/Dip ^b^	30 ^1^	(25.9%) ^1^	35 ^2^	(27.1%) ^2^	65 ^3^	(26.5%) ^3^
	University	33 ^1^	(28.4%) ^1^	35 ^2^	(27.1%) ^2^	68 ^3^	(27.8%) ^3^
HH ^c^ inc ^d^ (pa) ^e^	No income	9	(7.7%)	7	(5.4%)	16	(6.5%)
	Pension	5	(4.3%)	5	(3.8%)	10	(4.0%)
	AUD 1–51,999	32	(27.4%)	43	(33.1%)	75	(30.4%)
	AUD 52,000–103,999	28	(23.9%)	27	(20.8%)	55	(22.3%)
	AUD > 104,000	22	(18.8%)	17	(13.1%)	39	(15.8%)
	Do not know ^f^	21	(17.9%)	31	(23.8%)	52	(21.1%)
Living arrgmt ^g^	Live alone	15	(12.8%)	23 ^4^	(17.8%) ^4^	38 ^5^	(15.4%) ^5^
	PR ^h^/Spouse only	69	(59.0%)	59 ^4^	(45.7%) ^4^	128 ^5^	(52.0%) ^5^
	Single/PR ^h^ (CH ^i^)	18	(15.4%)	25^4^	(19.4%) ^4^	43 ^5^	(17.5%) ^5^
	Parent/other	15	(12.8%)	22^4^	(17.1%) ^4^	37 ^5^	(15.0%) ^5^
Smoking status	Yes	44	(37.6%)	28	(21.5%)	72	(29.1%)
	No	73	(62.4%)	102	(78.5%)	175	(70.9%)

^a^ Cert = certificate. ^b^ Dip = diploma. ^c^ HH = household. ^d^ inc = income. ^e^ pa = per annum. ^f^ Combined with do not wish to answer. ^g^ arrgmt = arrangements. ^h^ PR = partner/partnered. ^i^ CH = children. ^1^ Education *n* = 116 males. ^2^ Education *n* = 129 females. ^3^ Total *n* = 245 for education. ^4^ Living arrangements *n* = 129 females. ^5^ Total *n* = 246 for living arrangements.

**Table 2 nutrients-13-04130-t002:** Descriptive statistics of anthropometric characteristics, chronic health conditions and categories of BMI in the study sample of Australian adults.

	Female		Male		Total	
	*n* = 130		*n* = 117		*n* = 247	
Characteristics	Median	(IQR)	Median	(IQR)	Median	(IQR) *
Height (cm)	163 *	(6.5) *	176.8 *	(6.6) *	169.5 *	(9.5) *
Weight (kg)	70.2	(22.6)	88.9	(18.7)	79.7	(24.3)
Waist circumference (cm)	84.9	(19.9)	98.9	(19.3)	92.0	(22.2)
BMI (kg/m^2^)	*n*	(100%)	*n*	(100%)	*n*	(100%)
Normal	22 ^1^	(18.8%) ^1^	50	(39.1%)	72 ^2^	(29.4%) ^2^
Overweight	56 ^1^	(47.9%) ^1^	41	(32.0%)	97 ^2^	(39.6%) ^2^
Obese	39 ^1^	(33.3%) ^1^	37	(28.9%)	76 ^2^	(31.0%) ^2^
Number and proportions of chronic health conditions						
0	43	(33.1%)	48 ^3^	(41.7%) ^3^	91 ^4^	(36.8%) ^4^
1	45	(34.6%)	35 ^3^	(30.4%) ^3^	80 ^4^	(32.4%) ^4^
2	22	(16.9%)	17 ^3^	(14.8%) ^3^	39 ^4^	(15.8%) ^4^
3	14	(10.8%)	5 ^3^	5(4.3%) ^3^	19 ^4^	(7.7%) ^4^
4	4	(3.1%)	9 ^3^	9(7.7%) ^3^	13 ^4^	(5.3%) ^4^
5	2	(1.5%)	1 ^3^	1(0.9%)^3^	3 ^4^	(1.2%) ^4^
Individual diagnosed Chronic health conditions						
Circulatory conditions	36	(27.6%)	31 ^5^	(26.0%) ^5^	67 ^6^	(27.3%) ^6^
Chronic kidney or renal disease	1	(0.8%)	2 ^5^	(1.7%) ^5^	3 ^6^	(1.2%) ^6^
Diabetes (type 1, type 2 or gestational)	9	(6.9%)	8 ^5^	(7.0%) ^5^	17 ^6^	(6.9%) ^6^
Overweight or obesity	27	(20.8%)	25 ^5^	(21.7%) ^5^	52 ^6^	(21.2%) ^6^
Cancer (any)	6	(4.6%)	10 ^5^	(8.7%) ^5^	16 ^6^	(6.5%) ^6^
Chronic mental health conditions ^a^	24	(18.5%)	10 ^5^	(8.7%) ^5^	34 ^6^	(13.9%) ^6^
Musculoskeletal conditions ^b^	32	(24.6%)	26 ^5^	(22.6%) ^5^	58 ^6^	(23.7%) ^6^
Respiratory conditions ^c^	22	(16.9%)	14 ^5^	(12.2%) ^5^	36 ^6^	(14.7%) ^6^
None of the above	49	(33.1%)	43 ^5^	(42.6%) ^5^	92 ^6^	(37.6%) ^6^

^a^ Including: anxiety, depression, schizophrenia or other mental chronic health condition. ^b^ Including: back problems, osteoarthritis, rheumatoid arthritis. ^c^ Including: asthma, COPD or any other lung condition. * Mean and standard deviation are reported for height. ^1^ BMI *n* = 128 for females. ^2^ Total *n* = 245 for BMI. ^3^ Number and proportion of chronic conditions *n* = 115 for males. ^4^ Number and proportion of chronic conditions *n* = 245. ^5^ Individual diagnosed chronic health conditions *n* = 115 for males. ^6^ Individual diagnosed chronic health conditions *n* = 245.

**Table 3 nutrients-13-04130-t003:** Total Australian Recommended Food Score (ARFS), subscales, reference ranges, mean and standard deviation for a sample Australian adults categorised by gender.

		TOTAL		Male		Female	
ARFS Subscales	Reference Range	Mean	(SD)	Mean	(SD)	Mean	(SD)
		(*n* = 247)		(*n* = 117)		(*n* = 130)	
Total	0–73	34.5	(9.0)	33.4	(8.9)	35.5	(9.2)
Vegetables	0–21	13.5	(4.2)	13	(4.2)	13.9	(4.1)
Fruit	0–12	5.5	(2.7)	5.3	(2.7)	5.7	(2.7)
Meat	0–7	3.1	(1.5)	3.2	(1.5)	3.0	(1.4)
Meat alternatives	0–6	2.2	(1.3)	2.0	(1.2)	2.3	(1.3)
Grains	0–13	5.0	(2.2)	4.9	(2.2)	5.1	(2.1)
Dairy	0–11	3.9	(1.8)	3.7	(1.7)	4.1	(1.9)
Extras	0–1	0.8	(0.8)	0.9	(0.8)	0.8	(0.7)
Water	0–2	0.6	(0.5)	0.6	(0.5)	0.7	(0.5)

**Table 4 nutrients-13-04130-t004:** Multivariate linear regression demonstrating the association between socio demographic variables and total ARFS.

		Multivariate		(R^2^ 0.077, *p* ≤ 0.001)	
Characteristics	Levels	β Coefficient	SE ^a^	95% CI ^b^	*p*
Rurality	Major cities (*n* = 57)	Reference Category		-	-
	Regional/remote (*n* = 190)	−0.4	1.4	(−3.0, 2.3)	0.790
Age (years)	18–30 (*n* = 44)	Reference Category		-	-
	31–50 (*n* = 40)	5.4	2.6	(0.3, 10.4)	0.037 *
	51–70 (*n* = 125)	4.4	2.1	(0.3, 8.5)	0.035 *
	71 > (*n* = 38)	6.5	2.5	(1.6, 11.4)	0.010 *
Living arrgmt ^c^	Alone (*n* = 38)	Reference Category		-	-
	PR ^d^/Spouse only (*n* = 128)	5.2	1.6	(2.0, 8.4)	<0.002 *
	Single/PR ^d^ (CH) ^e^ (*n* = 43)	5.6	2.1	(1.4, 9.8)	0.008 *
	Parents/other (*n* = 37)	5.8	2.4	(1.1, 10.5)	0.016 *
Number of chronic health conditions	Continuous variable (*n* = 245)	−1.4	0.5	(−2.3, −0.4)	0.004 *

**^a^** SE = standard Error. ^b^ CI = confidence interval. ^c^ arrgmt = arrangements. ^d^ PR = partner/partnered. ^e^ CH = children. * = significance < *p* 0.20.

## Data Availability

The data presented in this study may be available on request from the Chief Investigator Associate Professor Leanne Brown. The data are not publicly available due to privacy.

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
