# Peer review of "A Comparison of Diet Quality in a Sample of Rural and Urban Australian Adults"

_nutrients, 2021, doi:10.3390/nu13114130_

Round 1
Reviewer 1 Report
Dietary habits are very important in multifactorial diseases. Authors assessed a relationship between diet quality and rural and urban populations in Australia. The manuscript is very interesting and well prepared. I only would like to convince the authors to continuous collecting data for enhance the sensitivity their research.
Author Response
I would like to thankyou on behalf of the team, for both your suggestions and time in reviewing our manuscript.
Kind regards
Rebekah Pullen
Reviewer 2 Report
The manuscript, "A comparison of diet quality in a sample of rural and urban Australian adults" is well-written and would be of interest to readers of nutrients.
My main concern is the low sample size (n=247) for this type of study and data; The authors acknowledge the low sample size in the limitations section. A couple of possibilities to address this: does the entire sample 638, significantly differ in any of the key variables being examined here? If there are no key differences, that would increase the confidence that the the smaller sample was reflective of the larger (though still small sample). The conclusions could be written in a way that using words such as indicate, suggest, (and not "demonstrated"). A power calculation could be done to show that with a sample size of 247, you'd be able to detect of difference of XX.
Other general comments:
‘Changing Health Actions At Rural Gala Events in 20 minutes’ (CHAARGE20) – attending galas – how is this group of attendees different than the typical rural individual, is this group representative of a rural population. Transportation to get there, mobility/well-enough to travel, wealthier? More connected? AND how would this impact results?
What’s the importance of score differences – at what difference in scores is there a health impact? Does it matter if it’s a 5-point increase between 33 to 38 or 46 to 51, that is if you’re in an excellent category does it a make a difference (health outcome wise) to move to the outstanding category. How does a change in score translate into better health outcomes? Some mention (beyond increased nutrient intakes) of what the improvements in scores mean when differences are around 5 or decreased scores of -1.5 would be useful.
Specific Comments (minor):
Specific comments:
lines 128-130, useful to classify the ARFS score into four groups: ‘needs work’ (<33), ‘getting there’ (33–38), ‘excellent’ (39–46), or ’outstanding’ (47+). please insert the total possible range of ARFS score
Lines 144-145, please give the specific categories for each of the variables and how they were collapsed (i.e., what are the collapsed categories)
Lines 144-149 were the questions to capture these variables open ended or did they have response categories. For example, chronic conditions listed a set of conditions or was it a write-in. For Chronic Conditions, did the question ask, have you been diagnosed by a health professional or was it more open such as what diseases do you have – this information is in the Table Footnote, perhaps it would be better placed in the methods section (Move the information as footnotes in the Tables to the Text in methods, authors can decide).
Lines 168-169 please provide the range of p values for the Shapiro Wilk test. No need to provide the pvalues for all variables, just the range and perhaps indicating the variable with the highest and the variable with the lowest pvalues. With such a low sample size, this will provide useful information. Perhaps state that the data were not transformed
Line 169 Delete the word “Therefore” statistically, it is not connected to the sentence before.
Lines 210-212 Table 2 Footnote: obesity classification – delete in footnote, authors do not use all the classifications and the classifications used by the authors is already defined in methods.
Lines 212-213 Including: Anxiety, depression, schizophrenia or other mental chronic health condition. These mental health conditions are quite different. For analysis, I recommend analyzing the data with anxiety and depression separately then analyze the data with all mental health conditions as listed here to see if / how it changes your results.
Table 4: Some of the Cis seem off if the results are significantly different – please check Cis and significance for age categories.
Lines 229-230 could be deleted if question regarding chronic conditions was asked in methods
Lines 330-332 “Another potential limitation of the present study was the high prevalence (21.1%) of respondents not wishing to disclose income. This could have produced misleading results and ignored the contribution of income to diet quality.” A possible method for addressing this limitation is to conduct the analysis -for all analyses in this manuscript - including only those whom disclosed income to see if you see the same trend (even if there is no significance).
